# Glacier surge activity over Svalbard from 1992 to 2025 interpreted using heritage satellite radar missions and Sentinel-1

Tazio Strozzi<sup>1</sup>, Erik Schytt Mannerfelt<sup>2,3</sup>, Oliver Cartus<sup>1</sup>, Maurizio Santoro<sup>1</sup>, Thomas Schellenberger<sup>2</sup>, Andreas Kääb<sup>2</sup>

<sup>1</sup>Gamma Remote Sensing, Gümligen, 3073, Switzerland

<sup>2</sup>Department of Geosciences, University of Oslo, Oslo, 0316, Norway

<sup>3</sup>Arctic Geology, The University Centre in Svalbard, Longyearbyen, 9170, Norway

Correspondence to: Tazio Strozzi (strozzi@gamma-rs.ch)

Abstract. Based on massive processing of heritage radar data from the satellite missions ERS-1/2, JERS-1, ENVISAT 10 ASAR, ALOS PALSAR and Radarsat-2, and in combination with data from the current Sentinel-1 and ALOS-2 PALSAR-2 missions, we compiled a ~30-year time series of radar backscatter over Svalbard. We exploited this data to detect glacier surges by using changes in backscatter as an indicator of increased or decreased surge-related crevassing. In this way, we reconstructed an as consistent as possible time series of surge activity on Svalbard for 1992 to 2025. We recorded 24 surgetype events during the pre Sentinel-1 period 1992-2014 (23 years) and 34 surge-type events during the post Sentinel-1 period 15 2015-2025 (11 years). This time series shows an approximately threefold increase in surges since 2015, from an average of about one surge per year to more than three surges per year. We show that this increase is unlikely to be explained alone by the better resolution, coverage and quality of the Sentinel-1 data compared to the data from the earlier SAR heritage missions. Simulation results indicate that the observed increase is extremely unlikely to be attributed to random perturbations in surge cyclicity, and instead suggest the influence of an external forcing mechanism. The number of surges during the 20 recent decade seems high, but due to uncertainties in historical records, it remains unclear whether this frequency is exceptional or if earlier decades were unusually quiet. The cause of the observed threefold increase in surge activity also remains uncertain, given our incomplete understanding of surge initiation in relation to climate variability and non-climatic surge controls.

## 1 Introduction

Glacier surging refers to strongly enhanced ice flow speeds over time-periods of months to years (Jiskoot, 2011; Truffer et al., 2021). Knowing where and when glaciers show surge-type flow instabilities is important for a number of scientific and applied reasons. The mechanisms of glacier surging and the conditions leading to it are still incompletely understood (Jiskoot et al., 2000; Murray et al., 2003a; Benn et al., 2019; Thøgersen et al., 2024) and questions arise whether and how climate change could impact surge initiation, frequency and magnitude, and therefore on the response of glaciers to climate change.

Glacier surges are identified and mapped using a number of (often combined) indicators such as looped moraines, specific




landforms in the glacier forefield, exceptional and major glacier advance, exceptional crevassing, sheared-off glacier tributaries or particular patterns of elevation and surface velocity change (Kääb et al., 2023).

Satellite Synthetic Aperture Radar (SAR) data have long been used for the observation of glacier surges, as they are weatherindependent and can be acquired at night. In particular, SAR interferometry (InSAR) and SAR offset-tracking have been used extensively to determine surge velocities (Luckman et al., 2002; Murray et al, 2003; McMillan et al, 2014; Schellenberger et al., 2015; Strozzi et al, 2017). More recently, Leclercq et al. (2021) introduced a method to detect surgetype glacier flow instabilities through the change in Sentinel-1 C-band backscatter across repeated SAR images. This radar backscatter procedure reveal major surge events characterised by extensive crevassing, but not slower, long-term instabilities that do not not cause significant changes on the surface. Using this approach, Kääb et al. (2023) mapped 26 surge-type events over Svalbard in the period 2017-22, a frequency of surge events that appears to be larger than in previous published inventories or studies (e.g. Farnsworth et al., 2016, recompiled in Harcourt et al., 2025b). The question therefore arises as to whether the increasing number of detected surge events is related to changing environmental or climatic conditions over Svalbard or simply due to improved observation capacity by satellites in the Sentinel era since 2015. As surge detection and definition considerably depend on the method used (Kääb et al., 2023), we avoid in this study combining results from multiple different methods and studies. Rather, in order to answer the above research question and extend the backscatterbased inventory back in time before the Sentinel-1 based inventory, we considered heritage satellite radar missions in the period 1992-2015. We first describe the data from the different SAR missions and its processing for this study. Then, we present method specificities of surge detection and results from applying it to the individual radar data sets. Finally, we discuss the combination of these mission-specific surge detections to a time series and draw some overall conclusions.

# 50 2 SAR data and processing

Our study uses data from five heritage satellite radar missions: ERS-1/2, JERS-1, ENVISAT ASAR, ALOS PALSAR and RADARSAT-2. In addition, we have updated the Sentinel-1 based surge inventory 2017-2022 from Kääb et al. (2023) and used ALOS-2 PALSAR-2 data for further comparison and confirmation.

## 2.1 ERS-1/2

ERS was ESA's first Earth-observation satellite programme using a polar orbit (ERS Overview: https://earth.esa.int/eogateway/missions/ers/description/, last accessed 26 September 2025). ERS-1 was launched in 1991 and operational until 2000 with various mission phases using 3-day, 35-day and 168 day repeat cycles. ERS-2 was launched in 1995, had a repeat cycle of 35 days and was deactivated in 2011. Both ERS-1 and ERS-2 were operated in a 1-day difference between 1995 and 2000. The two satellites provided a large dataset of long-term observations that supported a range of applications. However, the need for direct read-out of the SAR telemetry necessitated a global network of ground stations, which has led to widely varying numbers of acquisitions depending on geographic area and year.

We processed the full archive of ERS-1/2 C-band SAR data over Svalbard acquired between mission launch in 1991 and the end of the mission in 2011 at VV polarisation. The European Space Agency (ESA) Heritage Mission Programme provided access to 17'333 images in Single Look Complex (SLC) format. A fully automated processing chain was implemented based on the GAMMA SAR and InSAR software (Wegmüller and Werner, 1997) to process the SLC data to Radiometric Terrain-Corrected (RTC) level in equiangular projection with a pixel posting of 0.000666667° in Easting and 0.0002222222° in Northing (ca. 25 m). A 20 m LiDAR Digital Elevation Model (DEM) available from the Norwegian Polar Institute (Norwegian Polar Data Centre: https://data.npolar.no/, last accessed 26 September 2025) was used for ortho-rectification and topographic correction. The processing was organized on a per-orbit basis, i.e., all images acquired from the same orbit were processed simultaneously. For each SAR image the processing included:

- 1. replacing the orbit state vectors with DELFT precision orbits (Scharroo and Visser, 1998);
- 2. 2x9 multi-looking in range and azimuth, respectively, to obtain Multi-Look Intensity images (MLI) with a ground range pixel posting of about 25 m;
- 3. calculation of transformation functions, so called geocoding look-up tables (Wegmüller, 1998), for resampling between slant-range and map geometry based on the available orbit information and DEM;
  - 4. calculation of maps of the local incidence angle and layover/shadow masks;
  - 5. refinement of the geocoding look-up tables because of the low accuracy of the orbit information based on an iterative refinement procedure implemented with a simulated backscatter image calculated from the DEM<sup>1</sup>;
  - 6. pixel area normalization as with Frey et al. (2013) to obtain RTC gamma0 backscatter images;
- 7. resampling to the target map geometry in geographic spatial reference system based on the refined geocoding look-up tables;
  - 8. mosaicking of all geocoded backscatter images from the same orbit;
  - 9. storage of the geocoded backscatter images as 1° x 1° tiles accompanied by local incidence angle maps, layover/shadow maps, and CARD4L compliant XML files (Normalised Radar Backscatter: https://ceos.org/ard/files/PFS/NRB/v5.5/CARD4L-PFS\_NRB\_v5.5.pdf, last accessed 26 September 2025) containing summaries of the most relevant information about the imagery and pre-processing, including information about the geocoding accuracy.

<sup>1</sup> Despite use of the DELFT vectors we often found offsets in the order of 100 m between the real MLI and the simulated backscatter images, especially in the range direction. For frames without topography, for which refinement was not possible, the offsets of the neighbouring images acquired along same orbit were considered.

## 2.2 JERS-1

85

90

Throughout the seven years of operation between 1992 and 1998 (About Japanese Earth Resources Satellite "FUYO-1" (JERS-1): https://global.jaxa.jp/projects/sat/jers1/index.html, last accessed 26 September 2025), JERS-1 was able acquire an almost complete global coverage, i.e., each area worldwide was covered at least once. In contrast to more recent spaceborne

100

- SAR missions, however, no systematic acquisition plan had been defined for JERS-1, which is why land areas were not covered systematically on an annual basis but rather in the frame of mission phases dedicated to cover certain areas once or repeatedly from consecutive JERS-1 orbital cycles; note that JERS-1 was flown in a 44 days repeat orbit.
  - GAMMA processed the global archive of JERS-1 L-band SAR data acquired at HH polarisation between mission launch in February 1992 and the end of the mission in October 1998 to RTC level (Global 25 m Resolution JERS-1 SAR Mosaic: https://www.eorc.jaxa.jp/ALOS/en/dataset/pdf/DatasetDescription\_JERS-1\_Mosaic\_ver200.pdf, last accessed 26 September 2025). The Japanese Space Agency (JAXA) provided access to 478'447 images in Level 0 raw format that were available in JAXA's archives. The processing was essentially carried out using the same method as described for ERS-1/2, with the
- 1. focussing of the raw data to SLC format including filtering for Radio Frequency Interference (RFI) effects (Natsuaki et al., 2017), antenna gain corrections, as well as corrections for the noise floor;
  - 2. 2x7 multi-looking in range and azimuth, respectively;
  - calculation of the geocoding look-up tables based on the available orbit state vectors information and the global ALOS-1 PRISM DEM AW3D30 with 1 arc-second resolution (ALOS World 3D 30m: https://www.eorc.jaxa.jp/ALOS/en/dataset/aw3d30/aw3d30\_e.htm, last accessed 26 September 2025);
- 4. for areas without topography, refinement of the geocoding look-up tables because of the low accuracy of the JERS-1 orbit information (Shimada and Isoguchi, 2002) with a 20 m resolution ALOS-2 PALSAR-2 Fine-Beam Dual-polarisation (FBD) L-band HH polarisation backscatter dataset released by JAXA in the form of global mosaics (Shimada and Ohtaki, 2010; Shimada et al., 2014).

## 2.3 ENVISAT ASAR

following differences:

ENVISAT, an advanced polar-orbiting Earth observation satellite which provided measurements of the atmosphere, ocean, land, and ice, was launched by ESA in March 2002 and operated until the unexpected loss of contact in April 2012 (Envisat: https://earth.esa.int/eogateway/missions/envisat/description, last accessed 26 September 2025). All the ENVISAT Advanced Synthetic Aperture Radar (ASAR) images acquired in the Image Mode (IM) and the Wide Swath Mode (WSM), and available on ESA's G-POD (Cossu et al., 2008), were processed to a common reference system in geographic map projection and a pixel size of 150 m in northing and easting (Santoro et al., 2015). The pixel size was selected to be close to the native resolution of the mode with the poorer spatial resolution, i.e. WSM. The ASAR database consists of co-registered images of the radar backscatter, radiometrically calibrated and speckle filtered. The co-registration accuracy is at sub-pixel level (Santoro et al., 2015). North of 60° N and over large parts of Europe, more than 300 observations per pixel and per year are common. For approximately 3% of the area mapped, corresponding to northern Greenland, Svalbard and the Arctic isles of Russia, more than 1,000 observations per pixel and per year are available.

## 2.4 ALOS PALSAR and ALOS-2 PALSAR-2

The Phased Array type L-band Synthetic Aperture Radar (PALSAR) was an active microwave sensor using L-band frequency on board of the Advanced Land Observing Satellite (ALOS) launched in 2006 and operated until 2011 with a repeat cycle of 46 days (ALOS: https://www.eorc.jaxa.jp/ALOS/en/alos/a1\_about\_e.htm, last accessed 26 September 2025).

An observation strategy was developed by JAXA to provide spatially and temporally consistent, multi-seasonal global coverage, on a repetitive basis (Rosenqvist et al., 2004). However, the number of observations was usually limited to a few per year. The PALSAR-2 sensor aboard the ALOS-2 satellite is an L-band SAR launched in 2014 (ALOS-2: https://www.eorc.jaxa.jp/ALOS-2/en/about/palsar2.htm, last accessed 26 September 2025). Also for ALOS-2 PALSAR-2 JAXA developed an observation strategy to provide spatially and temporally consistent, multi-seasonal global coverage, on a repetitive basis; note that ALOS-2 is flown in a 14 days repeat orbit. ALOS-2 PALSAR-2 has different modes of acquisition providing users with more detailed data than ALOS PALSAR. The observation repetition frequency of ALOS-2 PALSAR-2 has been improved by expanding the satellite's observation range by approximately three times compared to ALOS PALSAR.

From the ALOS PALSAR and ALOS-2 PALSAR-2 images, JAXA generates annual, global SAR mosaics with a pixel size 140 (Global 25 Resolution PALSAR-2/PALSAR m m Mosaic: https://www.eorc.jaxa.jp/ALOS/en/dataset/pdf/DatasetDescription PALSAR2 Mosaic ver230a.pdf, accessed 26 September 2025). The mosaics are created by assembling long paths of SAR backscatter images observed through JAXA's Global Basic Observation Scenario. Correction of geometric distortions specific to SAR (ortho-rectification) and topographic effects on image intensity (radiometric slope correction) are applied to accommodate image analysis without the need for 145 further processing. The ALOS PALSAR annual mosaics are available for years between 2007 and 2010 and were last reprocessed by JAXA in 2022 according to the new approach and product format defined for ALOS-2 PALSAR-2 (Version 2). The new mosaics have improved geometric performance and radiometric balancing between adjacent paths to make the products seamless. The ALOS-2 PALSAR-2 mosaics are available since 2015. Each mosaic contains observations from the same year only, thus being affected by data gaps in regions where acquisitions were not succesfull. The mosaics are provided 150 as gamma0 backscatter, in geographical (lat/long) coordinates with a pixel spacing of 0.8 arc seconds (approximately 25 meters at the Equator). The datasets are available in HH and HV polarisation and also accessible in Google Earth Engine.

#### **2.5 RADARSAT-2**


The RADARSAT-2 satellite was launched in 2007 with a C-band sensor on board and has a repetition cycle of 24 days. RADARSAT-2 is a commercial radar imaging satellite and the data used in this study over Svalbard were acquired between 2011 and 2016 by NSC/KSAT under Norwegian–Canadian RADARSAT agreements (Schellenberger et al., 2015). We processed a series of 39 SLC products in Wide mode (nominal swath width of 150 km and maximal spatial resolution of 25 m) between 2011 and 2015 and 52 SLC products in Wide Fine mode (nominal swath width of 150 km and maximal spatial

resolution of 8 m) between 2012 and 2016 at HH polarisation. The fully automated processing chain followed essentially the same method as described above for ERS-1/2. However, a 100 m version of the Norwegian Polar Institute DEM (Norwegian Polar Data Centre: https://data.npolar.no/, last accessed 26 September 2025) was used for ortho-rectification and topographic correction and the SLC data images were resampled in the UTM zone 33N projection with a posting of 50 m x 50 m.

## 2.6 Sentinel-1




In addition to extending the radar-based surge inventory back in time using SAR heritage missions, we updated the Sentinel-1 based inventory over 2017-2022 from Kääb et al. (2023) forward to the year 2025, using the same method as described in the original publication (Leclercq et al., 2021). Since over Svalbard Sentinel-1 Interferometric Wide-Swath data (IW, swath width of 250 km and spatial resolution of about 20 meters) are not regularly available before 2017, we also used for this study Extended Wide-Swath data (EW, swath width of 400 km and spatial resolution of about 100 meters) to expand the Sentinel-1-based inventory to 2015. As for IW, the EW amplitude data are available in Google Earth Engine, which we use for our backscatter data analysis.

## 170 3 Surge detection

Leclercq et al. (2021) introduced a method to detect surge-type glacier flow instabilities through the change in backscatter that they cause in repeat satellite SAR images. The method was developed based on Sentinel-1 C-band backscatter data between consecutive years. First, aggregated images of maximum backscatter values for each pixel location over the 3-month winter period from January to March, when glaciers in the Northern hemisphere typically show little other backscatter change due to cold and dry conditions, were created. Then, the normalized difference between the two aggregated maximum images (so called Normalized Difference Index – NDI) was calculated to visualize change in backscatter and to identify surge activity, because surge-induced changes in surface roughness lead typically to an easily discernible increase or decrease in radar backscatter. In order to minimize the effect of variation of topographic effects, the analysis is preferably performed with images taken from the same nominal orbit.

Due to Sentinel-1's systematic acquisition strategy, a consistently large number of observations are available over Svalbard every winter for the same nominal orbits. For the heritage satellite radar missions, however, there are not many repeated winter observations for the same orbital track in consecutive years. The strict application of the methodology developed for Sentinel-1 means that many areas on Svalbard are very often not covered with a backscattering intensity change image in consecutive years to identify glacier surges. A number of modifications and different parameterisations of this basic concept have been therefore implemented depending on the available data (e.g. mission, ascending/descending orbits, number of annual acquisitions, polarisation, frequency, etc.). Below, we provide a description of the number and quality of images available over Svalbard from historical missions, introduce the approach considered for calculating the backscattering

intensity change images and show examples of glacier surge detection using the various sensors considered in this study, highlighting the specific characteristics of the different datasets.

## 190 3.1 ERS-1/2



In order to illustrate the coverage of Svalbard every winter season with ERS-1/2 data, we computed the mosaics of the average backscatter intensity for descending and ascending orbits using data acquired between November and April (Figures S1 and S2 in the Supplement). In Table 1 we summarize qualitatively the coverage of Svalbard with winter ERS-1/2 mosaics between 1992 and 2011. In order to get wall-to-wall coverage of all Svalbard, we need to consider differences for time scales longer than one year. To detect annual change in radar backscatter we computed therefore ratio images between years per season and ascending/descending orbits for all combinations of years up to 10 years time difference. To search for change in backscatter and to eventually identify surge activity we nevertheless mainly considered the years highlighted with a vertical arrow in Table 1.

Two examples of surge detection based on ERS-1/2 C-band SAR intensity change are presented in Figures 1 and 2. In Figure 1 we show the increase in backscattering intensity from 1999 to 2002 for Ingerbreen (Sund et al., 2009). The backscattering intensity change image is scaled from -10 dB in blue to +10 dB in red, with the interval between -3.33 dB and + 3.33 dB shown in white. With this representation the large increase in crevasses during the initiation of the surge is well depicted in red. In Figure 2 we show the decrease in backscattering intensity from 2003 to 2008 for Perseibreen (Dowdeswell and Benham, 2003). In this case the decrease in crevasses at the end of the surge is well depicted in blue.

Table 1: Coverage of Svalbard with winter (from November to April) ERS-1/2 mosaics between 1992 and 2011 for descending (D) and ascending (A) orbits: F is full covered, P is partly covered, I is incompletely covered and - is no data (see Figures S1 and S2). The vertical arrows indicate the years mainly considered for surge detection. Note that years in the legend refer to the end of the winter, i.e., 1996 means e.g. November 1995 to April 1996.

|   | 1992 | 1993 | 1994 | 1995 | 1996     | 1997 | 1998     | 1999 | 2000 | 2001 | 2002 | 2003 | 2004 | 2005 | 2006 | 2007 | 2008 | 2009 | 2010 | 2011 |
|---|------|------|------|------|----------|------|----------|------|------|------|------|------|------|------|------|------|------|------|------|------|
| D | P    | -    | -    | P    | F        | P    | F        | F    | P    | P    | F    | F    | P    | P    | I    | I    | F    | I    | I    | P    |
| A | P    | -    | I    | -    | P        | I    | P        | P    | I    | I    | I    | I    | I    | I    | I    | -    | F    | -    | -    | P    |
|   | 1    |      |      |      | <b>↑</b> |      | <b>↑</b> | 1    |      |      | 1    | 1    |      |      |      |      | 1    |      |      | 1    |

Figure 1: Example for detecting glacier surges from ERS-1/2 backscattering intensity change images for Ingerbreen. The change image in dB (left) is calculated as the ratio between the mosaics of winter backscatter intensity from 2002 (right) and 1999 (centre).

Figure 2: Example for detecting glacier surges from ERS-1/2 backscattering intensity change images for Perseibreen. The change image in dB (left) is calculated as the ratio between the mosaics of winter backscatter intensity from 2008 (right) and 2003 (centre).

## 3.2 JERS-1





Figure S3 shows annual mosaics of the average backscatter intensity for images acquired between November and April since 1993 and until 1998. For consistency with previous studies (Kääb et al., 2023), the data are grouped by hydrological year, i.e. the winter of 1993 means autumn 1992 to spring 1993. Large areas of Svalbard are not covered in 1993, 1996 and 1997. For surge detection, we therefore only calculated ratio images for all combinations of the years 1994, 1995 and 1998. Two examples of surge detection based on JERS-1 L-band SAR intensity change are presented in Figures 3 and 4. In Figure 3 we show the increase in backscattering intensity from 1994 to 1998 for Fridtjovbreen (Murray et al., 2003b). The backscattering intensity change image is scaled from -5 dB in blue to +5 dB in red, with the interval between -1.67 dB and +1.67 dB shown in white. In this case, the large increase in crevasses during the initiation of this well-studied surge is well depicted in red. The lower part of Figure 3 also shows a change in backscatter intensity in 1994 along two orbits. Due to the small number of available JERS-1 observations, it is not possible to mitigate these effects. Other disadvantages of JERS-1 for surge detection are the L-band penetration into ice and firn and the lower resolution, which make the identification of surges more difficult than using C-band images, as other events than surges can also produce similar patterns of backscatter changes. In addition, an out-of-date DEM had to be used for terrain correction and the resulting correction artefacts may introduce apparent changes (e.g. at much changed calving fronts). Finally, some of the JERS-1 tiles show errors in the calibration in the form of bright stripes, which are due to erroneous information on the automatic gain control that could not be corrected for in the processing. As a result, more care is needed for surge identification with JERS-1 compared to ERS-1/2 as small/weak surges are harder to detect or are likely to remain undetected. Nevertheless, as shown in Figure 4, the increase in backscattering intensity from 1994 to 1998 for Chydeniusbreen, a glacier with strong evidence for surging (Jiskoot et al., 2000) that has, however, not yet been studied in detail, is well discernible.

Figure 3: Example for detecting glacier surges from JERS-1 backscattering intensity change images for Fridtjovbreen. The change image in dB (left) is calculated as the ratio between the mosaics of winter backscatter intensity 1998 (right) and from 1994 (centre).

Figure 4: Example for detecting glacier surges from JERS-1 backscattering intensity change images for Chydeniusbreen. The change image in dB (left) is calculated as the ratio between the mosaics of winter backscatter intensity from 1998 (right) and 1995 (centre).

## 3.3 ENVISAT ASAR

We considered for historical surge detection over Svalbard the ENVISAT ASAR images acquired between winter 2003 and winter 2010. Given the large number of ENVISAT ASAR observations available over Svalbard, monthly averages of all ENVISAT ASAR images were first calculated at 150 m resolution for all images covering a given area, regardless of orbital direction and frames. This implies that data with different incidence angles are averaged. The number of observations on which the time average is based is also recorded. Over Svalbard, almost only WSM data are available. In the following, mosaics of the ENVISAT ASAR backscatter intensity from January to March of every year between 2003 and 2010 were computed. The whole of Svalbard is well covered with these winter data and only the period from January to March 2005 is missing (Figure S4). Apart from the variations in the incidence angle, which are particularly visible in the ocean and in mountainous areas, the signal in this averaged data appears to be quite clear. Finally, winter backscatter change images were calculated between subsequent years.

Two examples of surge detection based on ENVISAT ASAR C-band SAR winter-to-winter intensity change are presented in Figure 5 and 6. In Figure 5 we show the increase in backscattering intensity from 2009 to 2010 for the Nathorstbreen glacier system, including Dobrowolskibreen, Polakkbreen and Zawadzkibreen (Sund et al., 2014). The backscattering intensity change image is scaled from -5 dB in blue to +5 dB in red, with the interval between -1.67 dB and +1.67 dB shown in white.



With this representation the large increase in crevasses during the surge is clearly visible in red. In Figure 6 we show the decrease in backscattering intensity from 2004 to 2006 for Tunabreen (Flink et al., 2015). In this case the decrease in crevasses at the end of the surge is well depicted in blue. The ENVISAT ASAR backscatter intensity resulting from the averaging of a large number of observations at 150 m resolution with different orbits and angles of incidence is rather smooth compared to that of ERS-1/2 resulting from the averaging of a small number of images at higher spatial resolution from the same orbit direction (Figures 1 and 2). However, backscattering intensity changes can be equally well depicted to detect glacier surges.

Figure 5: Example for detecting glacier surges from ENVISAT ASAR backscattering intensity change images for the Nathorstbreen glacier system, including Dobrowolskibreen, Polakkbreen and Zawadzkibreen. The change image in dB (left) is calculated as the ratio between the mosaics of winter backscatter intensity from 2010 (right) and 2009 (centre).

Figure 6: Example for detecting glacier surges from ENVISAT ASAR backscattering intensity change images for Tunabreen. The change image in dB (left) is calculated as the ratio between the mosaics of winter backscatter intensity from 2006 (right) and 2004 (centre).

### 3.4 ALOS PALSAR

The ALOS PALSAR mosaics provide good coverage of the whole of Svalbard (Figure S5) and the signal in this averaged data appears to be quite clear. Backscatter change images were thus calculated between subsequent years. However, considering the fact that the mosaics include data acquired throughout the whole year, these are only taken into account to confirm possible surges detected using data from the other satellite missions.

To illustrate the detection of surges using ALOS PALSAR L-Band data, we consider annual mosaics spatially averaged to a pixel size of 100 m. In Figure 7 we show the increase in backscattering intensity from 2007 to 2008 for Comfortlessbreen (Sund and Eiken, 2010). The backscattering intensity change image is scaled from -5 dB in blue to +5 dB in red, with the

interval between -1.67 dB and +1.67 dB shown in white. With this representation the large increase in crevasses during the surge is well depicted in red. However, as the images are averaged over the whole year, there are also changes in the backscatter in the regions around the glacier. In Figure 8 we show the increase in backscattering intensity from 2009 to 2010 for the Nathorstbreen glacier system, including Dobrowolskibreen, Polakkbreen and Zawadzkibreen (Sund et al., 2014). Similarly to ENVISAT ASAR (Figure 5), with this representation the large increase in crevasses during the surge is well depicted in red. However, because of the higher native spatial resolution, the image products obtained from the ALOS PALSAR mosaics are sharper than those obtained from the averaged ENVISAT ASAR mosaics.

Figure 7: Example for detecting glacier surges from ALOS PALSAR backscattering intensity change images for Comfortlessbreen. The change image in dB (left) is calculated as the ratio between the mosaics of yearly backscatter intensity from 2008 (right) and 2007 (centre).

Figure 8: Example for detecting glacier surges from ALOS PALSAR backscattering intensity change images for the Nathorstbreen glacier system, including Dobrowolskibreen, Polakkbreen and Zawadzkibreen. The change image in dB (left) is calculated as the ratio between the mosaics of winter backscatter intensity from 2010 (right) and 2009 (centre).

#### 3.5 RADARSAT-2



As can be seen from the mosaics of average backscatter intensity for ascending and descending orbits for the period October to April (Figure S6), there are only a few repeated Radarsat-2 winter observations in consecutive years over Svalbard. The strict application of the methodology developed for Sentinel-1 (Leclercq et al., 2021) means that many areas on Svalbard are seldom covered with a backscatter intensity change image in consecutive years to identify glacier surges. To get wall-to-wall coverage over all Svalbard, we need therefore to consider differences for time scales longer than one year. The differences between years per winter seasons were computed for all combinations of years up to 4 years time difference. In addition, we

computed the differences in the backscatter intensity between ENVISAT in 2010 and RADARSAT-2 in 2012 and between ERS-2 in 2011 and RADARSAT-2 in 2012.

Examples of surge detection based on Radarsat-2 C-band SAR intensity change are presented in Figures 9 and 10. In Figure 9 we show the increase in backscattering intensity from 2013 to 2014 for the front of Stonebreen (Strozzi et al., 2017). The backscattering intensity change image is scaled from -10 dB in blue to +10 dB in red, with the interval between -3.33 dB and + 3.33 dB shown in white. With this representation the slight increase in crevasses at the very beginning of the surge is depicted in red. In Figure 10, we show the increase in backscattering intensity from 2015 to 2016 for Wahlenbergbreen (Sevestre et al., 2018). In this case the large increase in crevasses during the initiation of the surge is very well depicted in red.

Figure 9: Example for detecting glacier surges from Radarsat-2 backscattering intensity change images for Stonebreen. The change image in dB (left) is calculated as the ratio between the mosaics of winter backscatter intensity from 2014 (right) and 2013 (centre).

Figure 10: Example for detecting glacier surges from Radarsat-2 backscattering intensity change images for Wahlenbergbreen. The change image in dB (left) is calculated as the ratio between the mosaics of winter backscatter intensity from 2016 (right) and 2015 (centre).

#### 320 3.6 Sentinel-1


The catalogue of Sentinel-1 images used in Kääb et al. (2023) was expanded with winter-to-winter backscattering intensity differences for the years 2022–2023, 2023–2024 and 2024–2025 using current IW data. We also expanded the IW-based inventory back in time for 2015–2017 with data collected in EW swath mode. Here, we applied the same surge detection



methodology as for the IW cross-polarised data (Kääb et al., 2023), being aware that this coarser data could lead to less detailed detection of surges or to overlooking surges of small glaciers or with only limited backscatter changes.

Examples of surge detection based on Sentinel-1 C-band IW and EW NDI are presented in Figure 11. An increase in the backscatter-based difference between 2024 and 2025 can be observed for Doktorbreen, Paulabreen and Nordsysselbreen, while for Liestølbreen, Scheelebreen, Vallåkrabreen and Kvalbreen we observe a decrease of the backscatter-based difference. We find that all surges detected using IW data are also well detectable using EW and vice-versa, although IW data offer more spatial details. This gives us confidence that the 2015-2017 EW-based extension of the Sentinel-1 based inventory is fully reliable compared to the IW-based results.

Figure 11: Example for detecting glacier surges from Sentinel-1 NDI images from maximum backscatter winter 2024 – winter 2025 computed from IW (left) and EW (right) data for (from south-west to north-east) Liestølbreen, Doktorbreen, Scheelebreen, Vallåkrabreen, Paulabreen, Nordsysselbreen and Kvalbreen. Outlines from the Randolph Glacier Inventory 7.0 (RGI 7.0 Consortium, 2023) are shown in light blue.

#### 3.7 ALOS-2 PALSAR-2

The global ALOS-2 PALSAR-2 mosaics provide good coverage of the whole of Svalbard (Figure S7). An example of surge detection is presented in Figure 12. The increase in crevasses during the surge of Monacobreen is well depicted in white, but there are also changes in the backscatter in the regions around the glacier, because the ALOS-2 PALSAR-2 mosaic consists of annual averages. Quality and usefulness of the mosaics to detect surges depended on several factors such as time of image acquisition, data coverage, number of observations per year etc. Surge detection is therefore less systematic compared to Sentinel-1, but, as indicated in Figure 12, ALOS-2 PALSAR-2 might well complement the Sentinel-1 based detection.

Figure 12: Left: example for detecting glacier surge from ALOS-2 PALSAR-2 annual backscattering intensity change images 2018-2019 for Monacobreen. Right: comparison to Sentinel-1 IW winter NDI images from maximum backscatter winter 2018 – winter 2019. Outlines from the Randolph Glacier Inventory 7.0 (RGI 7.0 Consortium, 2023) are shown in light blue.

## 4 Surge catalogue





We mapped 18 surge-type events over Svalbard over the period 1992-2011 using ERS-1/2 SAR data (Table 2 and Figure S8). Most of these glaciers were already well studied in previous works (i.e., Monacobreen (Luckman et al., 2002), Fridtjovbreen (Murray et al., 2003b), Nordre Franklinbreen (Pohjola et al., 2011), Kroppbreen (Sund et al., 2009), Perseibreen (Dowdeswell and Benham, 2003), Ingerbreen (Sund et al., 2009), Skobreen (Sund et al., 2009), Tunabreen (Flink et al., 2015), Dobrowolskibreen (Sund et al., 2014), Comfortlessbreen (Sund and Eiken, 2010) and Blomstrandbreen (Sund and Eiken, 2010)) or indicated as surge type or with strong evidence for surging on previous catalogues (i.e., Stubendorffbreen (Farnsworth, et al. 2016), Midtbreen (Farnsworth et al. 2016), Chydeniusbreen (Jiskoot et al., 2000), Mendelejevbreen (Sund et al., 2009) and Vasilievbreen named Skilfonna in Sund et al. (2009)). There is only one glacier without a name (G020828E78825N in Table 2), which was not mentioned in previous inventories, that showed a very crevassed front in the early 1990s. We interpreted this signal as a surge or at least a frontal destabilisation. Austfonna Basin 3 already showed a slight frontal advance and a weak increase in crevasses in the winter of 2011, but based on other works (McMillan et al., 2014; Schellenberger et al., 2017) and signals from other sensors (ALOS PALSAR and Radarsat-2), we decided to set the onset of the surge to 2012.

We mapped 6 surge-type events over Svalbard over the period 1993-1998 using JERS-1 SAR data (Table 2 and Figure S9). No other glacier surges were detected with this sensor in the 1990s, in addition to those previously discussed with ERS-1/2. In a re-examination of the signals from glaciers that were indicated as surge type or with strong evidence for surging in the earlier catalogues of the 1990s (e.g. Hagen et al., 1993; Jiskoot et al., 2000), we found only one strong crevassed front over Osbornebreen in the winter of 1992. Here the signal was not previously considered a surge because in comparison to winter 1996 it resembles a frontal retreat, as has been observed on many other glaciers in recent years (Figure S13). However, for consistency with our work, we have added Osbornebreen to our catalogue.






We mapped 10 surge-type events over Svalbard over the period 2003-2010 using ENVISAT ASAR SAR data (Table 2 and 370 Figure S10). This mapping was performed in parallel to the analysis with ERS-1/2, and there are only three surges that were not recorded also with ERS-1/2, as there are no observations available with this sensor over South Spitsbergen after 2008 (Figures S1 and S2). The three glaciers Nathorstbreen, Zawadzkibreen and Polakkbreen are considered as one glacier system by Sund et al. (2014), but we count them as three units in our records according to Randolph Glacier Inventory 7.0 (RGI 7.0 Consortium, 2023).

We mapped 5 surge-type events over Syalbard over the period 2007-2010 using ALOS PALSAR data (Table 2 and Figure S12). Considering that the mosaics for ALOS PALSAR refer to the whole year and not only to the winter months, these data are only taken into account to confirm surges already detected with ERS-1/2 or ENVISAT, i.e. Comfortlessbreen (Sund and Eiken, 2010), the glacier system Nathorstbreen, Zawadzkibreen and Polakkbreen (Sund et al., 2014) and Blomstrandbreen (Sund and Eiken, 2010). In a re-examination of the signals from glaciers that were indicated as surge type or with strong 380 evidence for surging in the earlier catalogues of the 2000s (e.g. Sund et al., 2009), we found two glaciers with an unclear evidence of increased or decreased crevasses. According to Sund et al. (2009), the first year of surge observation for Ragna-Mariebreen was 2003 and we can observe a very weak increase in the backscattering of ERS-1/2 between 2004 and 2009 in the upper part of the glacier (Figure S13). For Bungebreen, which according to Sund et al. (2009) was first observed to surge in 2004, we also observe a slight increase in the backscattering of ERS-1/2 in the upper part of the glacier between 2004 and 2009 (Figure S13). However, neither glacier showed a clear frontal advance and, as we are not sure whether they are to be interpreted as a glacier surge, we have not included them in our list. A slight advance of the front of Bungebreen is later visible in the Sentinel-1 data, but without any noticable backscatter change. Therefore, Bungebreen was neither included in the inventory of Kääb et al. (2023).

We mapped 6 surge-type events over Svalbard over the period 2012- 2015 using Radarsat-2 SAR data (Table 2). The surges detected based on Radarsat-2 C-band SAR are presented in Figure S12, where we show the increase or decrease in backscattering intensity change between different years. Three of these surges have already been well studied in previous work, namely Austfonna Basin 3 (McMillan et al., 2014), Stonebreen (Strozzi et al., 2017) and Austfonna Basin 2 (Schellenberger et al., 2017), and Wahlenbergbreen has already been reported by Kääb et al. (2023). Only Aavatsmarkbreen and Bjørlykkebreen (a tributary of Lilliehöökbreen) started their surge before 2017, the year in which Kääb et al. (2023) started the study of recent glacier surges using Sentinel-1 data.

Kääb et al. (2023) mapped 26 surge-type events over Svalbard in the period 2017-2022 using Sentinel-1. Completing the Sentinel-1 analysis with old EW images before 2017 and with new IW acquisitions from 2023 onwards, 38 surge-type events were mapped over Svalbard using Sentinel-1 in the period 2015-2025 (Table 3), i.e. 12 more than for 2017-2022. Of these, Aavatsmarkbreen and Bjørlykkebreen are the two glaciers that were added before 2017, while Nordsysselbreen, Borebreen, Sefströmbreen, Doktorbreen, Etonbreen, Deltabreen, Johansenbreen, Petermannbreen (which were all also identified by Mannerfelt et al., 2025), Crollbreen and Davisbreen (which are new) are the 10 glaciers that were added after 2022.

Regarding the current situation, an important limitation of this approach is its difficulty to identify the start of new, slow

surges or confirm when existing surges have stopped from just one year of data. Only another year of data will reveal the corresponding trends in backscatter.

In addition to the glacier name, the Global Land Ice Measurements from Space Identification (GLIMS\_ID) and the RGI\_ID (RGI 7.0 Consortium, 2023) of the surge-like events interpreted over Svalbard using satellite radar data, we also indicate in Tables 2 and 3 the start and end year in which we observe an increase and decrease in backscatter, respectively. To characterise the duration of the surge more precisely, we interpreted the images of backscatter intensity and the images of intensity change from different sensors and years together. Since the annual records of the historical radar images are not complete, the list of surge-type events interpreted in the period 1992-2015 in Table 2 contains more approximate information than the list based on Sentinel-1 in Table 3. Based on the additional EW data before 2017 and on recent IW data, we were also able to update some surge information contained in the original Kääb et al. (2023) inventory, for instance concerning surge start and end years, and by adding the last year of strongly enhanced backscatter (before backscatter reduction).

Table 2: List of surge-type events interpreted over Svalbard over the period 1992-2015 using heritage satellite radar missions (E: 415 ERS-1/2, J: JERS-1, A: ENVISAT ASAR, P: ALOS PALSAR and R: RADARSAT-2). We also include the start and end year in which we observe increasing or decreasing backscatter, respectively, or indication if the start happened before 1992 (= 1000). To be consistent with the nomenclature of Sentinel-1, in the legend of Table 2 the year refer to the end of the winter, i.e., 1995 means the winter season 1994 to 1995.

| ID | GLIMS_ID       | RGI_ID                      | Lon      | Lat      | Surge_start | Surge_end | Name                 | Sensor |
|----|----------------|-----------------------------|----------|----------|-------------|-----------|----------------------|--------|
| 00 | G013139E78668N | RGI2000-v7.0-G-<br>07-00308 | 13.12553 | 78.65570 | 1000        | 1992      | Osbornebreen         | -      |
| 01 | G020828E78825N | RGI2000-v7.0-G-<br>07-01167 | 20.87410 | 78.82259 | 1000        | 1998      | NULL                 | E/J    |
| 02 | G012697E79319N | RGI2000-v7.0-G-<br>07-00054 | 12.62240 | 79.35666 | 1992        | 1998      | Monacobreen          | E/J    |
| 03 | G017040E78981N | RGI2000-v7.0-G-<br>07-01071 | 17.14306 | 79.00709 | 1994        | 1999      | Stubendorffbreen     | E/J    |
| 04 | G016346E79525N | RGI2000-v7.0-G-<br>07-01102 | 16.54762 | 79.51513 | 1994        | 1999      | Midtbreen            | E/J    |
| 05 | G014442E77835N | RGI2000-v7.0-G-<br>07-00651 | 14.42868 | 77.83081 | 1995        | 1999      | Fridtjovbreen        | E/J    |
| 06 | G017693E79137N | RGI2000-v7.0-G-<br>07-01074 | 17.77668 | 79.16570 | 1996        | 2004      | Chydeniusbreen       | E/J    |
| 07 | G016557E76915N | RGI2000-v7.0-G-<br>07-00783 | 16.57598 | 76.92757 | 1999        | 2003      | Mendelejevbreen      | Е      |
| 08 | G020321E80038N | RGI2000-v7.0-G-<br>07-01343 | 20.22978 | 80.05299 | 1999        | 2004      | Nordre Franklinbreen | Е      |
| 09 | G017401E77894N | RGI2000-v7.0-G-<br>07-00963 | 17.40816 | 77.89775 | 2000        | 2005      | Kroppbreen           | Е      |
| 10 | G017400E77460N | RGI2000-v7.0-G-<br>07-00760 | 17.39924 | 77.48342 | 2001        | 2006      | Perseibreen          | E/A    |
| 11 | G018231E77758N | RGI2000-v7.0-G-<br>07-00973 | 18.28992 | 77.74233 | 2002        | 2004      | Ingerbreen           | E/A    |

| 12 | G017183E77703N | RGI2000-v7.0-G-<br>07-00746 | 17.22685 | 77.71341 | 2003 | 2007 | Skobreen          | E/A   |
|----|----------------|-----------------------------|----------|----------|------|------|-------------------|-------|
| 13 | G017497E78572N | RGI2000-v7.0-G-<br>07-00502 | 17.79310 | 78.54826 | 2004 | 2006 | Tunabreen         | E/A   |
| 14 | G016634E77343N | RGI2000-v7.0-G-<br>07-00912 | 16.77240 | 77.35627 | 2004 | 2008 | Dobrowolskibreen  | E/A   |
| 15 | G016898E76901N | RGI2000-v7.0-G-<br>07-00780 | 16.89817 | 76.89127 | 2007 | 2010 | Vasilievbreen     | E/A   |
| 16 | G012220E78757N | RGI2000-v7.0-G-<br>07-00281 | 12.08800 | 78.77236 | 2008 | 2011 | Comfortlessbreen  | E/A/P |
| 17 | G016633E77290N | RGI2000-v7.0-G-<br>07-00913 | 16.90377 | 77.31700 | 2008 | 2017 | Nathorstbreen     | A/P   |
| 18 | G015857E77338N | RGI2000-v7.0-G-<br>07-00899 | 15.82066 | 77.33716 | 2009 | 2017 | Zawadzkibreen     | A/P   |
| 19 | G016187E77275N | RGI2000-v7.0-G-<br>07-00902 | 16.32112 | 77.30436 | 2009 | 2017 | Polakkbreen       | A/P   |
| 20 | G012468E79076N | RGI2000-v7.0-G-<br>07-00246 | 12.37570 | 79.06692 | 2010 | 2015 | Blomstrandbreen   | E/P   |
| 21 | G024340E79634N | RGI2000-v7.0-G-<br>07-01383 | 24.32109 | 79.58774 | 2012 | 2020 | Austfonna Basin 3 | E/R   |
| 22 | G023608E77828N | RGI2000-v7.0-G-<br>07-01531 | 23.58579 | 77.80146 | 2012 | 2022 | Stonebreen        | R     |
| 23 | G012307E78697N | RGI2000-v7.0-G-<br>07-00291 | 12.22124 | 78.69644 | 2014 | 2016 | Aavatsmarkbreen   | R     |
| 24 | G024396E79406N | RGI2000-v7.0-G-<br>07-01385 | 24.53876 | 79.36849 | 2015 | 2018 | Austfonna basin 2 | R     |
| 25 | G011860E79450N | RGI2000-v7.0-G-<br>07-01312 | 11.78058 | 79.43103 | 2015 | 2020 | Bjørlykkebreen    | R     |
| 26 | G013901E78579N | RGI2000-v7.0-G-<br>07-00354 | 13.88470 | 78.55515 | 2015 | 2020 | Wahlenbergbreen   | R     |
| -  | G017374E77800N | RGI2000-v7.0-G-<br>07-00945 | 77.79493 | 17.36387 | -    | -    | Ragna-Mariebreen  | -     |
| -  | G016163E76838N | RGI2000-v7.0-G-<br>07-00803 | 16.13832 | 76.84135 | -    | -    | Bungebreen        | -     |

Table 3: List of surge-type events interpreted over Svalbard over the period 2015-present using Sentinel-1 (S). We also include the start and end year in which we observe increasing or decreasing backscatter, respectively, or indication if the end is not yet observed in 2025 (=3000), respectively. Duplicates with the list of Table 2 for the period up to 2015 are indicated in italic.

| ID | GLIMS_ID       | RGI_ID                      | Lon   | Lat   | Surge_start | Surge_end | Name                                                | Sensor |
|----|----------------|-----------------------------|-------|-------|-------------|-----------|-----------------------------------------------------|--------|
| 1  | G011860E79450N | RGI2000-v7.0-G-<br>07-01312 | 11.92 | 79.37 | 2015        | 2020      | Bjørlykkebreen<br>(tributary of<br>Lilliehöökbreen) | S      |
| 2  | G012307E78697N | RGI2000-v7.0-G-<br>07-00291 | 12.07 | 78.71 | 2014        | 2016      | Aavatsmarkbreen                                     | S      |

| 3  | G012363E79148N           | RGI60-                      | 12.36 | 79.15 | 2021 | 2024 | Fjortende Julibreen                   | S |
|----|--------------------------|-----------------------------|-------|-------|------|------|---------------------------------------|---|
|    | G01 <b>2</b> 50527711011 | 07.01492                    | 12.50 | 75.10 |      |      | 1 1011011111011011                    |   |
| 4  | G012697E79319N           | RGI60-<br>07.01494          | 12.55 | 79.47 | 2017 | 2021 | Monacobreen (recent surge)            | S |
| 5  | G013139E78668N           | RGI60-<br>07.00482          | 13.2  | 78.61 | 2018 | 2022 | Osbornebreen                          | S |
| 6  | G013724E78479N           | RGI2000-v7.0-G-<br>07-01548 | 13.97 | 78.43 | 2022 | 3000 | Borebreen                             | S |
| 7  | G013901E78579N           | RGI60-<br>07.00465          | 14.06 | 78.5  | 2015 | 2020 | Wahlenbergbreen                       | S |
| 8  | G013819E78747N           | RGI2000-v7.0-G-<br>07-01219 | 14.1  | 78.73 | 2023 | 3000 | Sefströmbreen                         | S |
| 9  | G015037E77377N           | RGI60-<br>07.00228          | 14.8  | 77.45 | 2018 | 2020 | Recherchebreen                        | S |
| 10 | G015616E77394N           | RGI60-<br>07.00241          | 15.63 | 77.47 | 2016 | 2019 | Penckbreen                            | S |
| 11 | G016346E79525N           | RGI60-<br>07.00849          | 16.1  | 79.54 | 2021 | 2024 | Midtbreen<br>(recent surge)           | S |
| 12 | G016633E77290N           | RGI2000-v7.0-G-<br>07-00913 | 16.19 | 77.42 | 2008 | 2017 | Nathorstbreen                         | S |
| 13 | G016915E77433N           | RGI60-<br>07.01472          | 16.7  | 77.43 | 2021 | 3000 | Liestølbreen                          | S |
| 14 | G016885E77574N           | RGI2000-v7.0-G-<br>07-00917 | 16.73 | 77.54 | 2021 | 3000 | Doktorbreen                           | S |
| 15 | G016742E76846N           | RGI60-<br>07.00299          | 16.76 | 76.86 | 2017 | 2020 | Vasilievbreen                         | S |
| 16 | G016777E76955N           | RGI60-<br>07.00440          | 16.78 | 76.96 | 2017 | 2020 | Svalisbreen                           | S |
| 17 | G016964E77694N           | RGI60-<br>07.00283          | 16.97 | 77.73 | 2020 | 3000 | Scheelebreen                          | S |
| 18 | G017158E77876N           | RGI60-<br>07.00266          | 17.11 | 77.87 | 2021 | 3000 | Vallåkrabreen                         | S |
| 19 | G017122E77256N           | RGI2000-v7.0-G-<br>07-00769 | 17.17 | 77.24 | 2020 | 2024 | Davisbreen                            | S |
| 20 | G017180E77206N           | RGI2000-v7.0-G-<br>07-00770 | 17.23 | 77.2  | 2019 | 2023 | Crollbreen                            | S |
| 21 | G017096E77164N           | RGI60-<br>07.00250          | 17.24 | 77.15 | 2017 | 2019 | Ryggbreen (tributary to Markhambreen) | S |
| 22 | G017399E77693N           | RGI60-<br>07.01470          | 17.4  | 77.69 | 2022 | 3000 | Paulabreen                            | S |
| 23 | G017497E78572N           | RGI60-<br>07.01458          | 17.43 | 78.5  | 2016 | 2019 | Tunabreen (recent surge)              | S |
| 24 | G017697E77678N           | RGI60-                      | 17.53 | 77.6  | 2015 | 2020 | Morsjnevbreen /                       | S |

|    |                | 07.00296                    |        |        |      |      | Strongbreen                                                 |   |
|----|----------------|-----------------------------|--------|--------|------|------|-------------------------------------------------------------|---|
| 25 | G017769E77848N | RGI2000-v7.0-G-<br>07-00967 | 17.93  | 77.88  | 2023 | 3000 | Nordsysselbreen                                             | S |
| 26 | G018031E77579N | RGI60-<br>07.00293          | 17.96  | 77.56  | 2020 | 2024 | Kvalbreen                                                   | S |
| 27 | G018098E77802N | RGI60-<br>07.00276          | 18.21  | 77.84  | 2016 | 2021 | Arnesenbreen (two surge phases)                             | S |
| 28 | G018419E78498N | RGI2000-v7.0-G-<br>07-01171 | 18.71  | 78.49  | 2020 | 2024 | Petermannbreen                                              | S |
| 29 | G018607E78547N | RGI2000-v7.0-G-<br>07-01170 | 18.72  | 78.53  | 2020 | 2024 | Johansenbreen                                               | S |
| 30 | G018042E78675N | RGI60-<br>07.01506          | 18.92  | 78.59  | 2015 | 2021 | Negribreen                                                  | S |
| 31 | G020098E78757N | RGI60-<br>07.00892          | 20.12  | 78.69  | 2019 | 2021 | Sonklarbreen                                                | S |
| 32 | G020757E78746N | RGI60-<br>07.00897          | 20.72  | 78.74  | 2016 | 2019 | Ganskijbreen                                                | S |
| 33 | G021502E79897N | RGI60-<br>07.00042          | 21.52  | 79.85  | 2018 | 2020 | Bodleybreen (outlet)                                        | S |
| 34 | G023608E77828N | RGI60-<br>07.01554          | 24.08  | 77.74  | 2012 | 2022 | Stonebreen; first<br>frontal instabilty,<br>then full surge | S |
| 35 | G024396E79406N | RGI60-<br>07.00026          | 24.48  | 79.36  | 2015 | 2018 | Austfonna basin 2                                           | S |
| 36 | G024340E79634N | RGI60-<br>07.00027          | 25.18  | 79.48  | 2012 | 2020 | Austfonna basin 3                                           | S |
| 37 | G023294E77648N | RGI2000-v7.0-G-<br>07-01466 | 23.17  | 77.69  | 2021 | 3000 | Deltabreen                                                  | S |
| 38 | G022972E79705N | RGI2000-v7.0-G-<br>07-01386 | 22.878 | 79.683 | 2024 | 3000 | Etonbreen                                                   | S |

## **5 Discussion**

We mapped over Svalbard 21 surge-type events in the period 1992-2011 with ERS-1/2 SAR, JERS-1 SAR, ENVISAT ASAR and ALOS PALSAR data and 6 surge-type events over the period 2012-2015 using Radarsat-2 SAR data (Table 2). With Sentinel-1, 38 surge-type events were mapped in the period 2015-2025 (Table 3). Eliminating duplicates between the two lists (italic in Table 3), we recorded 24 surge-type events initiated in the pre-Sentinel-1 period 1992-2014 (23 years; on average 1.0 surge/year) and 34 surge-type events initiated in the Sentinel-1 period 2015-2025 (11 years; on average 3.1 surges/year), which corresponds to about a threefold increase in surge events in the last period. More specifically, we recorded 5 surge-type events initiated before 1994, 10 surge-type events initiated in the 10 years between 1995 and 2005, 9

surge-type events initiated in the 10 years between 2005 and 2014, and 34 surge-type events initiated in the 11 years between 2015 and 2025, see graphical representation in Figure 13.

Figure 13: Surge events interpreted over Svalbard in the period 1992-2025 using ERS-1/2 SAR, JERS-1 SAR, ENVISAT ASAR, 435 ALOS PALSAR, Radarsat-2 and Sentinel-1 data listed by start date. Green is before 1994, yellow is 1995-2005, orange is 2005-2014 and red is 2015-2025.

## 5.1 Revisiting the Sentinel-1 based surges using ALOS-2 PALSAR-2

In order to investigate if the strong increase in surge frequency during the Sentinel-1 era is due to the better resolution and coverage compared to the heritage missions, we revisited the Sentinel-1 based surges using the ALOS-2 PALSAR-2 data. As described above, surges are typically more difficult to detect in the yearly ALOS-2 PALSAR-2 mosaics due to the varying seasonal cover, inclusion of summer data, lower resolution, and the L-band of the sensor. We view thus this comparison as a worst-case test whether the Sentinel-1 detected surges would also be detectable using the heritage missions, being well aware that this assumption is only coarse.

From the 38 surges detected using Sentinel-1 for 2015-2025, seven could not be detected using 2015-2023 ALOS-2

PALSAR-2 yearly mosaics. However, three of these started after 2023 and one ended in 2016, so that we cannot expect to see the final phase of crevasse closing as well in ALOS-2 PALSAR-2 data as we see it in Sentinel-1 data. From the remaining three, one surge (Bodleybreen) is also uncertain from the Sentinel-1 data, and the surges of Fjortende Julibreen and Petermannbreen show only weak backscatter changes in Sentinel-1 data. An additional five surges (Crollbreen, Davisbreen, Doktorbreen, Borebreen, Midtbreen) are detectable using ALOS-2 PALSAR-2 mosaics; however, their signatures are ambiguous and would likely have gone undetected without prior knowledge. In total, thus, from the 34 surges during the ALOS-2 PALSAR-2 phase, 26 surges are well detectable in the L-band data. Of these, 23 started in the period

2015-2023 (8 years), corresponding to a frequency of 2.9 surges/year. Based on this qualitative uncertainty assessment we suggest that an almost threefold increase of surge activity on Svalbard since about 2015 is in fact likely. A strictly quantitative uncertainty assessment of this number is, however, not possible with the method employed in this study.

## 5.2 Is surge frequency increasing stochastically?

Two reasons, or their combination, for this increase in surge activity seem possible. First, an interference pattern resulting from overlay of the individual surge cycle frequencies could lead to variations of the average number of surges by year, i.e. it could be a stochastic coincidence that surge activity increased. Kääb et al. (2023) did a similar investigation, but on global scale and in the spatial domain rather than in the temporal. They concluded that the current global clustering of glacier surges is very likely not a random interference. A geographical distribution of the surge events over Svalbard in the different time periods is shown in Figure 14. The surge events appear to be evenly distributed over the entire archipelago, regardless of the start dates, see also Figure S14.

Figure 14: Surge events interpreted over Svalbard in the period 1992-2025 using ERS-1/2 SAR, JERS-1 SAR, ENVISAT ASAR, ALOS PALSAR, Radarsat-2 and Sentinel-1 data listed by start date. Green is before 1994, yellow is 1995-2005, orange is 2005-2014 and red is 2015-2025.





To further test whether the observed increase from about one to about three surges per year could arise by chance under classical assumptions of surge dynamics (i.e., cyclical and intrinsically driven recurring events), we simulated a distribution of surge cycles as periodic events within a large feature space of possible configurations. We modeled surges as cyclic events with return times drawn from a normal distribution (mean 50–300 years, standard deviation 10–200 years), clamped to a minimum of 10 years because no faster return time has been observed in Svalbard, and assigned random phase offsets to each glacier. All combinations of these parameters were tested, and Monte Carlo simulations were run for each combination (Robert and Casella, 2004). We recorded the number of surges in 10-year windows and aggregated results across parameter sets that produced an overall mean frequency near the historical baseline (i.e., ~15±2.5 surges in 10 years; c.f the subsequent historical baseline discussion at Section 5.4). The simulation is conceptually similar to Approximate Bayesian Computation (Csilléry et al., 2010), except we focus on the outcome of the filtered runs instead of the posterior distribution of the parameters. The analysis was designed to assess plausibility across a wide feature space rather than estimate true probabilities; the percentages are rather a measure of the compatibility of a hypothesis for the given feature space.

Across the nearly 7 million simulations, the probability (i.e., compatibility with the feature space) of reaching more than 3 surges per year under random-phase interference was vanishingly small (~0.03%), even when exploring this broad feature space (Figure 15). In contrast, more than 2 surges per year occurred in ~13% of all simulations, indicating that moderate clustering can occur by chance, but the observed threefold increase is highly unlikely. Since the pre-1990s baseline surge frequency can be underestimated due to information bias, and hence in reality be higher than about 1.5 surges per year, we also tested a reverse scenario. Assuming that about 3 surges per year is the true baseline, how likely is it to observe the low frequency of the 1990s? The answer is the same: random-phase interference cannot explain such variability (Figure 15).

Figure 15: Cumulative probability functions of the momentary frequency of surge occurrence under different baseline surge frequencies due to constructive or destructive interference within the wide tested feature space, assuming that all surges have an independent (random) phase. The plots show that the probability of exceedence is directly tied to the baseline surge frequency; for example, more than 30 surges per decade will only consistently occur (>10% probability) with a baseline frequency of 20 surges per decade or more, and fewer than 10 will only consistently occur with a baseline of 15 surges per decade or less.



# 5.3 Is surge frequency increasing due to external forcing?

A second potential reason for the increase in Svalbard surge activity since about 2015 is an external forcing to surge frequency, e.g., through climatic forcing. To simulate scenarios of partial to full surge cycle synchronization, the analysis was re-run with a varying number of glaciers being assigned a phase of zero, i.e. they start surging at the same time. Partial or full phase-synchronization (surges set to start at the same time) can lead to periods of 3 surges per year or more (Figure 16), but the phase synchronization event must have occurred recently (within one or two mean return times) to remain highly plausible. These simulations strongly suggest that the recent surge frequency is not random and likely reflects a real change in system dynamics, i.e., partial or full phase synchronization.

As glacier surges exist in particular within a climate envelope constructed by temperature and precipitation variables (Sevestre and Benn, 2015), the impact of climate forcing on surge activity seems plausible. For our Svalbard results, this may point to regional-scale meteorological or climatic influences on surge timing. Detailed statistical analysis of this influence is, though, complicated by the incomplete understanding of surge initiation. Mannerfelt et al. (2025) showed that surges on Svalbard can initiate many years before they reach the phase of a fully developed surge and exhibit signs, in particular strong crevassing, that we use in this study to detect and time-stamp surge starts. This means the potential environmental changes behind a change in surge activity might well happen a decade before the observed changes in surge activity. As such, the positive climatic mass balances of Svalbard's glaciers between, roughly, 2005-2012 (Schmidt et al., 2023) could well have contributed to increased surge activity, perhaps in combination with a particularly negative mass balance year 2013 associated with high production of melt water. However, given the incomplete understanding of surge initiation we consider these connections not robust.








Figure 16: Exceedence probabilities of 30 surges per decade for the mean historical baseline surge frequency of 15 surges per decade within the wide tested feature space, tested over varying levels of phase synchronization. The plots reveal pulses of high momentary surge counts near the hypothetical synchronization event, before decorrelating to the baseline frequency over a number of average return period lengths. The analysis supports that many surges at once is much more easy to explain if some level of phase synchronization is introduced.

#### 5.4 Historical surge frequency

Comparison of our SAR backscatter-based inventory with the inventory compilation of Harcourt et al. (2025b) can be used to investigate surge frequencies on Svalbard, under the strong limitation that earlier surge observations are not systematic and expected to potentially contain substantial gaps in space and time. In particular, surges with observation years available in the Harcourt et al. (2025b) compilation are mostly from Liestøl (1993) and the current study. Liestøl (1993) indicates that "It is obvious that many more (glaciers) have surged but have not been observed". The inventory compilation of Harcourt et al. (2025b) shows that periods of high surge frequencies were at least seen around 1900 (~15 surges in ~10 years), around 1935 (~20 surges in ~10 years), and around 1965 (~15 surges in ~10 years). For the recent 11 years (i.e. Sentinel-1 era) we recorded ~30 surges. Taken into account the potential number of not-observed surges during the earlier periods suggests that the present number of surges in Svalbard is high but not far exceptional. The variability in the historical data may even suggest that increases or decreases have indeed happened before our study period as well, as was already suggested by Dowdeswell et al. (1995). The emphasis of our findings is on the more certain threefold difference between the last decade and the two decades before it.

Our new inventory compilation also contains some repeat surges. Within our own radar-based time series, three glaciers surged twice; Monacobreen (25 years difference), Midtbreen (27 years) and Tunabreen (12 years). Of the other surging glaciers identified in our study, around 20 were not observed to surge before our observation period, even if indirect geomorphologic indicators might exist (Farnsworth et al., 2016), and around 30 were directly observed surging before. Note that these numbers are approximate due to uncertainties in glacier names and other identifiers, as well as uncertainty as to whether different parts of the same glacier may have surged. For most glaciers with two dated surge observations available, the surge included in the current study is the second one. For glaciers with more than two dated observations, the last time difference is shorter than the first for six glaciers, roughly equal for four, and larger for one glacier (Crollbreen). The time differences between two surges range from ~140-150 years (Liestølbreen, Doktorbreen, Fridtjovbreen, Sefströmbreen) to ~10-20 years (Fjortende Julibreen, Paulabreen, Tunabreen). The number of glaciers with repeated, dated surge observations is, however, too small for reliable statistics, and we have to take into account considerable spatial and temporal gaps in observations dating back longer.

#### **6 Conclusions**

Unlocking archives of historical satellite missions has opened up new opportunities for environmental research. We have shown that the frequency of surge events in Svalbard has been increasing since 2015. This increase is unlikely to be








explained only by the better resolution, coverage and quality of the Sentinel-1 data compared to the data from the earlier SAR missions. In addition, our simulation results have indicated that the observed increase is extremely unlikely to be attributed to random perturbations in surge cyclicity, and instead suggest the influence of an external forcing mechanism. The number of surges observed for 2015-2025 is remarkable, but appears not to be drastically higher than for some earlier periods during the last century, notably around 1940, when accounting for potential information bias. In this respect, the still short interval of the time series compared to climate variability, the typical length of surge cycles on Svalbard and the incomplete understanding of surge triggering complicate the interpretation of the observed increase in surge frequency. Although thirty years of observations mark a climatic baseline, they merely represent the threshold for process understanding. As regular satellite radar missions only began in 1991, more evidences will be likely by gathering images in years to come. Alternatively, analyses of optical images from space, air or ground in combination with geomorphological analyses could contribute to an update of the catalogues of past surge events (Harcourt et al., 2025b). Within this framework, we recall that a fundamental uncertainty in this field stems from the lack of a clear definition of a glacier surge, as current definitions are dependent on the detection method used. Our backscatter approach, detecting mainly large surge-event, offers a distinct and potentially complementary alternative to other methods for surge characterization. Future work should also focus on classifying surges into at least two types (Sevestre et al., 2018); those that initiate at the terminus and propagate upglacier, and those that propagate downglacier. A comparison of these types could reveal differences between past and current surge behaviors.

While satellite archives offer invaluable insights into Svalbard's glacial surges, comparable data are sparser elsewhere. Due to its strategic location for monitoring the Arctic, its high latitude and proximity to Europe, Svalbard was frequently observed by satellites in polar orbits. Similar studies with heritage SAR missions can thus only be approximately carried out in other areas, such as the Russian Arctic, the Canadian Arctic or the Karakoram. To this end, the ESA Heritage Mission Programme shall process the corresponding volume of SLC data. Satellite SAR missions enable diverse applications beyond glacier surge monitoring, including support in global glacier mapping or glacier calving front delineation, e.g. using deep learning models which can process vast datasets (Maslov et al., 2025). Heritage satellite SAR data can also provide valuable benefits for these applications.

Our analysis of SAR data from multiple sensors operating at C- and L-band frequencies has yielded several key insights regarding surge detection and optimal acquisition strategies. First, we examined whether combining evidence from both frequency bands could enhance surge detection capabilities. Based on the available datasets, our findings suggest that such a confluence does not provide a measurable improvement under the observed conditions. Second, we considered whether a preferred acquisition strategy – encompassing frequency, polarization, repeat-pass cycle, and resolution – could be recommended. While the answer depends heavily on specific application requirements and environmental conditions, our analysis highlights the importance of balancing spatial resolution and temporal coverage. For instance, higher-resolution data may improve small glacier discrimination, whereas more frequent acquisitions could better capture highly dynamic processes.

## Acknowledgements

This research has been supported by ESA through Glaciers CCI (grant no. 4000109873/14/I-NB). We thank ESA's Heritage Space Programme for provision of the ERS-1/2 data archive. ERS-1/2 and ENVISAT ASAR data © ESA. The JERS-1 global mosaic dataset was generated by Gamma Remote Sensing, under special commission by JAXA EORC. JERS-1, ALOS PALSAR and ALOS-2 PALSAR-2 data © JAXA. RADARSAT-2 data were provided by NSC/KSAT under Norwegian—Canadian Radarsat agreements, © MDA. Sentinel-1 images are available from Copernicus.

## **Contributions**

T.Str. and A.K. conceptualised the study and wrote the manuscript. T.Str. implemented the methods and performed the analysis with heritage SAR missions. A.K. implemented the methods and performed the analysis using data from Sentinel-1 and ALOS-2 PALSAR-2. O.C. and M.S. performed the processing of the SAR data from ERS-1/2, JERS-1 and ENVISAT ASAR. E.M. performed the simulations of surge frequency and supported the analysis with heritage SAR missions. T.Sch. provided the Radarsat-2 data and supported their processing and interpretation. O.C., M.S., E.M. and T.Sch. reviewed the manuscript.

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
