# Peer review of "Glacier surge activity over Svalbard from 1992 to 2025 interpreted using heritage satellite radar missions and Sentinel-1"

_EGUsphere, 2025_

## Referee Comment (RC2)

**Review of egusphere-2025-5011: Glacier surge activity over Svalbard from 1992 to 2025 interpreted using heritage satellite radar missions and Sentinel-1**

**Summary**

This manuscript presents a comprehensive compilation of glacier surge events in Svalbard spanning more than 30 years, using data from five heritage satellite SAR missions (ERS-1/2, JERS-1, ENVISAT ASAR, ALOS PALSAR and Radarsat-2) and current satellite SAR missions (Sentinel-1 and ALOS-2 PALSAR-2). Building on the work of Kääb et al. (2023), the authors extend the observation period to 2015-2025 and reprocess the dataset following the methodology of Leclercq et al. (2021). The updated analysis largely confirms previous findings while identifying several additional surge events. The authors report a notably higher number of surges during the most recent decade compared to earlier periods since 1992. Through simulations, they argue that neither random phase interference nor purely external forcing can fully explain the observed variability in surge frequency. They suggest that periods of positive climatic mass balance in the mid-2000s, potentially combined with enhanced meltwater production in 2013, may have contributed to increased surge activity. However, given the incomplete understanding of surge initiation mechanisms, these links remain tentative. While the number of surges observed between 2015 and 2025 is remarkable, it does not appear unprecedented when accounting for potential observational biases in earlier periods, such as around the 1940s.

This is a well-written manuscript that aligns closely with the scope of The Cryosphere and provides valuable insights into glacier surge activity in Svalbard. The long-term perspective and consistent reprocessing of multi-mission SAR data are particularly strong aspects of the study. I have only minor comments and suggestions that may help further improve the clarity and consistency of the manuscript.

**Specific comments**

- L430: The phrase should read "between 1995 and 2004". Referring to the period between 1995 and 2005 would imply an 11-year interval.
- Figure 13
  The vertical line between 2024 and 2025 appears to indicate a decadal boundary. If so, this should be clarified. Otherwise, it seems inconsistent with the periodization already shown (-1994, 1995-2004, 2005-2014, 2015-2025). While not a major issue, it would improve clarity to adopt a fully consistent scheme in both manuscript and figures (e.g., -1995 / 1995-2005 / 2005-2015 / 2015-2025, or -1994 / 1995-2004 / 2005-2014 / 2015-2024 / 2025-present).

- Table 2 and 3
  - The ordering of rows is unclear. Table 2 appears to be sorted by ascending surge start date, whereas Table 3 seems to be ordered by longitude. For consistency, I suggest sorting both tables by surge start date.
  - Additionally, Tables 2 and 3 use different coordinate precisions for latitude and longitude. Is this due to the use of different DEMs? If so, this should be clarified in the table caption.
  - For table 3, it would also be helpful to indicate, for the italicized cases, which additional sensors observed the same surge events.
- Use of X-band SAR data

  Do the authors plan to include X-band SAR missions (e.g., TSX/TDX/PAZ, COSMO-SkyMed, or commercial missions such as ICEYE and Capella) in future analyses? While the manuscript notes that L-band does not provide a measurable improvement under the observed conditions, it would be interesting to briefly discuss whether X-band observations could offer a useful compromise. Collaboration with commercial providers may also increase temporal sampling of surge evolution, which could be valuable for future studies.

---

## Author Comment (AC1)

In the responses below the reviewers' comments are in black and our responses are in red.

Firstly, I want to thank the authors for completing the work for the community. It can be challenging and time-consuming to work with multiple historical satellite missions. As outlined in the manuscript, each dataset typically has distinct characteristics, and special considerations or care are often needed to ensure high-quality analysis results. I acknowledge such a huge effort associated with this work.

In this study, the authors examine five heritage satellite radar datasets and Sentinel-1 data to identify surge onset and termination in Svalbard using methods from Leclercq et al. (2021) and Kääb et al. (2023). The results agree well with existing records and have updated our knowledge of surging glaciers in Svalbard by adding a few more events and clarifying previous identifications. The study also shows that both C-band and L-band can be used to identify surge events based on radar backscatter changes, which could be useful information for future radar missions. The manuscript is well prepared, with detailed descriptions, and the discussion offers interesting validations and perspectives on surge frequency and external forcings. I enjoyed reading it.

The manuscript is ready to be accepted by TC, in my opinion, but if the authors have time, the following comments may be considered before publication.

We thank the reviewer for his positive comments and constructive suggestions regarding our manuscript. In the responses below, we address the suggestions made and explain the changes we will make to the manuscript.

- What is the availability of the data generated by this work, especially the mosaicked annual radar maps and the maps of NDI for each glacier? It would be good to guide the readers with a data availability section.

The data generated by this work are available on request from the corresponding author. The data is not publicly available, as this would require considerable effort in terms of preparation and documentation and goes far beyond the scope of our project. We will add a section on Data availability.
[The data generated by this work are available on request from the corresponding author.]

In addition, as indicated by the Copernicus Publications editorial support team, we will add a section on Declaration of Competing Interest.
[The authors declare that they have no conflict of interest.]

- L302-304: The ENVISAT—RADARSAT-2 and ERS-2—RADARSAT-2 analyses seem to be the only cross-platform comparisons throughout the study, but I can't find any results or discussion later in the manuscript. What is their performance in terms of surge detection?

Correct, we additionally computed the differences in the backscatter intensity between ENVISAT in 2010 and RADARSAT-2 in 2012 and between ERS-2 in 2011 and RADARSAT-2 in 2012. However, as these images did not reveal any other surge, they were lost during our analyses and the redaction of the manuscript. We thank the reviewer for this remark and will include in the revision of the supplement images of the differences in the backscatter intensity between ENVISAT in 2010

and RADARSAT-2 in 2012 for Blomstrandbreen and the Nathorstbreen glacier system (also including Zawadzkibreen and Polakkbreen) and an image of the differences in the backscatter intensity between ERS-2 in 2011 and RADARSAT-2 in 2012 for Blomstrandbreen. In addition, we will update the Sensor column in Table 2 by including RADARSAT-2 for Nathorstbreen, Zawadzkibreen, Polakkbreen and Blomstrandbreen and ENVISAT for Blomstrandbreen and we will include a short text describing these analyses in the manuscript. Please be aware that there were no observations available with ERS-1/2 over South Spitsbergen after 2008, including the Nathorstbreen glacier system.

[Figure]

Nathorstbreen_2012-2010    Blomstrandbreen_ 2012-2010    Blomstrandbreen_ 2012-2011

-10          dB          +10

*Backscattering intensity change images from ENVISAT in 2010 and RADARSAT-2 in 2012 for (left) the Nathorstbreen glacier system, also including Zawadzkibreen and Polakkbreen, and (middle) Blomstrandbreen. Backscattering intensity change image from ERS-2 in 2011 and RADARSAT-2 in 2012 for Blomstrandbreen (right). The name of the glacier and the dates of the yearly mosaics are indicated below each image.*

- L469: For one simulation, are all surges drawn from the same normal distribution? Or is each surge drawn from a different normal distribution with the mean and standard deviation randomly assigned? How many surge events are drawn?

All surges are drawn from the same normal distribution in each simulation. This distribution varies between simulations depending on the mean and standard deviation. We tested a uniform distribution of 50-1000 glaciers that actively take part in the statistics (i.e., "active surge-type glaciers"). To be more specific on this point, we will write in the revised version of the manuscript: "We modeled surges as cyclic events with 50-1000 active members and return times drawn from the same normal distribution depending on the mean and standard deviation (mean 50–300 years, standard deviation 10–200 years) [...]."

- L491-498: It would be helpful to provide more details for readers to understand the simulation. According to Figure 16, this analysis is only performed for the case with F = 15/decade, correct? How many surging glaciers are there in one run? Are there also a few million simulations aggregated into these results?

We indeed only present results for F = 15 /decade. There is a variable number of active glaciers (now clarified in the above comment); the runs represent the filtered members where the random baseline frequency is 15+-2.5 / decade, so the number highly varies. We start with 7 million results,

but after filtering, the number is about 2000 members. The plot shows exceedence probabilities for simulations of these ~2000 members. To be more specific on this point, we will write in the revised version of the manuscript: "Partial or full phase-synchronization (surges set to start at the same time) given the historical baseline can lead to periods of 3 surges per year or more (Figure 16), but [...]"

- Copyediting suggestions:
    - L244: … detection over Svalbard "using" the ENVISAT…?
      The sentence "We considered for historical surge detection over Svalbard the ENVISAT ASAR images acquired between winter 2003 and winter 2010." will be changed to "We considered the ENVISAT ASAR images acquired between winter 2003 and winter 2010 for historical surge detection over Svalbard."
    - Figure 11: Since the glaciers are not aligned along the southwest to northeast direction, it would be good to add labels of glacier names to each surging glacier for better identification with the caption.
      Agreed, labels of glacier names will be added to Figure 11.

---

## Author Comment (AC2)

In the responses below the reviewers' comments are in black and our responses are in red.

**Summary**

This manuscript presents a comprehensive compilation of glacier surge events in Svalbard spanning more than 30 years, using data from five heritage satellite SAR missions (ERS-1/2, JERS-1, ENVISAT ASAR, ALOS PALSAR and Radarsat-2) and current satellite SAR missions (Sentinel-1 and ALOS-2 PALSAR-2). Building on the work of Kääb et al. (2023), the authors extend the observation period to 2015-2025 and reprocess the dataset following the methodology of Leclercq et al. (2021). The updated analysis largely confirms previous findings while identifying several additional surge events. The authors report a notably higher number of surges during the most recent decade compared to earlier periods since 1992. Through simulations, they argue that neither random phase interference nor purely external forcing can fully explain the observed variability in surge frequency. They suggest that periods of positive climatic mass balance in the mid-2000s, potentially combined with enhanced meltwater production in 2013, may have contributed to increased surge activity. However, given the incomplete understanding of surge initiation mechanisms, these links remain tentative. While the number of surges observed between 2015 and 2025 is remarkable, it does not appear unprecedented when accounting for potential observational biases in earlier periods, such as around the 1940s.

This is a well-written manuscript that aligns closely with the scope of The Cryosphere and provides valuable insights into glacier surge activity in Svalbard. The long-term perspective and consistent reprocessing of multi-mission SAR data are particularly strong aspects of the study. I have only minor comments and suggestions that may help further improve the clarity and consistency of the manuscript.

We thank the reviewer for their positive comments and constructive suggestions regarding our manuscript. In the responses below, we address the suggestions made and explain the changes we will make to the manuscript.

**Specific comments**

• L430: The phrase should read "between 1995 and 2004". Referring to the period between 1995 and 2005 would imply an 11-year interval.

Thank you for pointing out this error. This sentence will be changed to "10 surge-type events initiated in the 10 years between 1995 and 2004".

• Figure 13
The vertical line between 2024 and 2025 appears to indicate a decadal boundary. If so, this should be clarified. Otherwise, it seems inconsistent with the periodization already shown (-1994, 1995-2004, 2005-2014, 2015-2025). While not a major issue, it would improve clarity to adopt a fully consistent scheme in both manuscript and figures (e.g., -1995 / 1995-2005 / 2005-2015 / 2015-2025, or -1994 / 1995-2004 / 2005-2014 / 2015-2024 / 2025-present).

We agree that adopting in the manuscript a decennial scheme such as -1994 / 1995-2004 / 2005-2014 /2015-2024 / 2025- would be more consistent. The bottom line is that in winter 2025 we did not detect any change in surge activity, so we simply updated in the manuscript the period 2015–2024 (10 years) to 2015–2025 (11 years). In fact, it is hard to tell from backscatter data if a surge

stopped in winter 2025, because we know only by winter 2026 if the 2025 backscatter signals were the last. To some extent the same is also true for the starting surges: with a first sign of backscatter increases we don't know yet for sure if the glacier enters a surge. This is a principle problem inherent to the method that it is perhaps worth to mention. In Figure 13, however, we also had to insert a ten-year line between 2024 and 2025 to ensure consistency.  Since you acknowledge that this is not a major issue, we will simply clarify in the caption to Figure 13 that the vertical lines indicate a ten-year boundary without any modification to the manuscript.

• Table 2 and 3
- The ordering of rows is unclear. Table 2 appears to be sorted by ascending  surge start date, whereas Table 3 seems to be ordered by longitude. For consistency, I suggest sorting both tables by surge start date.

Agreed, we will also sort Table 3 by surge start date.

- Additionally, Tables 2 and 3 use different coordinate precisions for latitude and longitude. Is this due to the use of different DEMs? If so, this should be clarified in the table caption.

We would like to thank the reviewer for pointing out the differences between Table 2 and Table 3. Table 3 was compiled on the basis of the supplementary material to Kääb et al. (2023, https://doi.org/10.1017/jog.2023.35), which is based on RGI v6.0 and uses a three digits precision for latitude and longitude. Table 2 was specifically prepared for this publication, is based on RGI v7.0 and uses a five digits precision for latitude and longitude. Therefore, there are indeed inconsistencies between the two tables, e.g.
- Table 2, Wahlenbergbreen, G013901E78579N, RGI2000-v7.0-G-07-00354, 13.88470, 78.55515.
- Table 3, Wahlenbergbreen G013901E78579N, RGI60-07.00465 14.063 78.504.
In the revision, we will consistently use RGI v7.0 for both Table 2 and Table 3 and apply the same precision for latitudes and longitudes.

- For table 3, it would also be helpful to indicate, for the italicized cases, which additional sensors observed the same surge events.

Agreed, in Table 3 we will indicate for the cases printed in italics which additional sensors observed the same surge events.

• Use of X-band SAR data
Do the authors plan to include X-band SAR missions (e.g., TSX/TDX/PAZ, COSMO-SkyMed, or commercial missions such as ICEYE and Capella) in future analyses? While the manuscript notes that L-band does not provide a measurable improvement under the observed conditions, it would be interesting to briefly discuss whether X-band observations could offer a useful compromise. Collaboration with commercial providers may also increase temporal sampling of surge evolution, which could be valuable for future studies.

In the past, we had access to satellite SAR data from X-band missions (e.g. TerraSAR-X and ICEYE) over Svalbard. These images are well suited for identifying surges based on changes in backscatter, but we have no specific plans for future analyses using these images. Nevertheless, we will expand the final paragraph of the conclusions by explicitly mentioning X-band data. For instance: "In this regard, X-band SAR missions such as TerraSAR-X, COSMO-SkyMed, ICEYE or Capella offer a good opportunity to detect local surges."